# Sarcasm SIGN: Interpreting Sarcasm with Sentiment Based Monolingual Machine Translation

## Abstract

Sarcasm is a form of speech in which speakers say the opposite of what they truly mean in order to convey a strong sentiment. In other words, *"Sarcasm is the giant chasm between what I say, and the person who doesn't get it."*. In this paper we present the novel task of sarcasm interpretation, defined as the generation of a non-sarcastic utterance conveying the same message as the original sarcastic one. We introduce a novel dataset of 3000 sarcastic tweets, each interpreted by five human judges. Addressing the task as monolingual machine translation (MT), we experiment with MT algorithms and evaluation measures. We then present SIGN: an MT based sarcasm interpretation algorithm that targets sentiment words, a defining element of textual sarcasm. We show that while the scores of n-gram based automatic measures are similar for all interpretation models, SIGN's interpretations are scored higher by humans for adequacy and sentiment polarity. We conclude with a discussion on future research directions for our new task.[1]

## 1 Introduction

Sarcasm is a sophisticated form of communication in which speakers convey their message in an indirect way. It is defined in the Merriam-Webster dictionary (Merriam-Webster, 1983) as the use of words that mean the opposite of what one would really want to say in order to insult someone, to show irritation, or to be funny. Considering this definition, it is not surprising to find frequent use of sarcastic language in opinionated user generated content, in environments such as Twitter, Facebook, Reddit and many more.

In textual communication, knowledge about the speaker's intent is necessary in order to fully understand and interpret sarcasm. Consider, for example, the sentence *"what a wonderful day"*. A literal analysis of this sentence demonstrates a positive experience, due to the use of the word *wonderful*. However, if we knew that the sentence was meant sarcastically, *wonderful* would turn into a word of a strong negative sentiment. In spoken language, sarcastic utterances are often accompanied by a certain tone of voice which points out the intent of the speaker, whereas in textual communication, sarcasm is inherently ambiguous, and its identification and interpretation may be challenging even for humans.

In this paper we present the novel task of interpretation of sarcastic utterances. We define the purpose of the interpretation task as the capability to generate a non-sarcastic utterance that captures the meaning behind the original sarcastic text.

Our work currently targets the Twitter domain since it is a medium in which sarcasm is prevalent, and it allows us to focus on the interpretation of tweets marked with the content tag #sarcasm. And so, for example, given the tweet *"how I love Mondays. #sarcasm"* we would like our system to generate interpretations such as *"how I hate Mondays"* or *"I really hate Mondays"*. In order to learn such interpretations, we constructed a parallel corpus of 3000 sarcastic tweets, each of which has five non-sarcastic interpretations (Section 3).

Our task is complex since sarcasm can be expressed in many forms, it is ambiguous in nature and its understanding may require world knowledge. Following are several examples taken from our corpus:

1. *loving life so much right now. #sarcasm*
2. *Way to go California! #sarcasm*

---

[1] Our dataset and code will be made publicly available.

3. *Great, a choice between two excellent candidates, Donald Trump or Hillary Clinton. #sarcasm*

In example (1) it is quite straightforward to see the exaggerated positive sentiment used in order to convey strong negative feelings. Examples (2) and (3), however, do not contain any excessive sentiment. Instead, previous knowledge is required if one wishes to fully understand and interpret what went wrong with California, or who Hillary Clinton and Donald Trump are.

Since sarcasm is a refined and indirect form of speech, its interpretation may be challenging for certain populations. For example, studies show that children with deafness, autism or Asperger's Syndrome struggle with non literal communication such as sarcastic language (Peterson et al., 2012; Kimhi, 2014). Moreover, since sarcasm transforms the polarity of an apparently positive or negative expression into its opposite, it poses a challenge for automatic systems for opinion mining, sentiment analysis and extractive summarization (Popescu et al., 2005; Pang and Lee, 2008; Wiebe et al., 2004). Extracting the honest meaning behind the sarcasm may alleviate such issues.

In order to design an automatic sarcasm interpretation system, we first rely on previous work in established similar tasks (section 2), particularly machine translation (MT), borrowing algorithms as well as evaluation measures. In section 4 we discuss the automatic evaluation measures we apply in our work and present human based measures for: (a) the *fluency* of a generated non-sarcastic utterance, (b) its *adequacy* as interpretation of the original sarcastic tweet's meaning, and (c) whether or not it captures the sentiment of the original tweet. Then, in section 5, we explore the performance of prominent phrase-based and neural MT systems on our task in development data experiments. We next present the *Sarcasm SIGN* (***S***arcasm ***S***entimental ***I***nterpretation ***G***e***N***erator, section 6), our novel MT based algorithm which puts a special emphasis on sentiment words. Lastly, in Section 7 we assess the performance of the various algorithms and show that while they perform similarly in terms of automatic MT evaluation, SIGN is superior according to the human measures. We conclude with a discussion on future research directions for our task, regarding both algorithms and evaluation.

## 2 Related Work

The use of irony and sarcasm has been well studied in the linguistics (Muecke, 1982; Stingfellow, 1994; Gibbs and Colston, 2007) and the psychology (Shamay-Tsoory et al., 2005; Peterson et al., 2012) literature. In computational work, the interest in sarcasm has dramatically increased over the past few years. This is probably due to factors such as the rapid growth in user generated content on the web, in which sarcasm is used excessively (Maynard et al., 2012; Kaplan and Haenlein, 2011; Bamman and Smith, 2015; Wang, 2013) and the challenge that sarcasm poses for opinion mining and sentiment analysis systems (Pang and Lee, 2008; Maynard and Greenwood, 2014). Despite this rising interest, and despite many works that deal with sarcasm identification (Tsur et al., 2010; Davidov et al., 2010; González-Ibánez et al., 2011; Riloff et al., 2013; Barbieri et al., 2014), to the best of our knowledge, *generation of sarcasm interpretations* has not been previously attempted.

Therefore, the following sections are dedicated to previous work from neighboring NLP fields which are relevant to our work: sarcasm detection, MT, paraphrasing and text summarization.

**Sarcasm Detection** Recent computational work on sarcasm revolves mainly around detection. Due to the large volume of detection work, we survey only several representative examples.

Tsur et al. (2010) and Davidov et al. (2010) presented a semi-supervised approach for detecting irony and sarcasm in product-reviews and tweets, where features are based on ironic speech patterns extracted from a labeled dataset. González-Ibánez et al. (2011) used lexical and pragmatic features, e.g. emojis and whether the utterance is a comment to another person, in order to train a classifier that distinguishes sarcastic utterances from tweets of positive and negative sentiment.

Riloff et al. (2013) observed that a certain type of sarcasm is characterized by a contrast between a positive sentiment and a negative situation. Consequently, they described a bootstrapping algorithm that learns distinctive phrases connected to negative situations along with a positive sentiment and used these phrases to train their classifier. Barbieri et al. (2014) avoided using word patterns and instead employed features such as the length and sentiment of the tweet, and the use of rare words.

Despite the differences between detection and

interpretation, this line of work is highly relevant to ours in terms of feature design. Moreover, it presents fundamental notions, such as the sentiment polarity of the sarcastic utterance and of its interpretation, that we adopt. Finally, when utterances are not marked for sarcasm as in the Twitter domain, or when these labels are not reliable, detection is a necessary step before interpretation.

**Machine Translation** We approach our task as one of monolingual MT, where we translate sarcastic English into non-sarcastic English. Therefore, our starting point is the application of MT techniques and evaluation measures. The three major approaches to MT are phrase based (Koehn et al., 2007), syntax based (Koehn et al., 2003) and the recent neural approach. For automatic MT evaluation, often an n-gram co-occurrence based scoring is performed in order to measure the lexical closeness between a candidate and a reference translations. Example measures are NIST (Doddington, 2002), METEOR (Denkowski and Lavie, 2011), and the widely used BLEU (Papineni et al., 2002), which represents *precision*: the fraction of n-grams from the machine generated translation that also appear in the human reference.

Here we employ the phrase based Moses system (Koehn et al., 2007) and an RNN-encoder-decoder architecture, based on Cho et al. (2014). Later we will show that these algorithms can be further improved and will explore the quality of the MT evaluation measures in the context of our task.

**Paraphrasing and Summarization** Tasks such as paraphrasing and summarization are often addressed as monolingual MT, and so they are close in nature to our task. Quirk et al. (2004) proposed a model of paraphrasing based on monolingual MT, and utilized alignment models used in the Moses translation system (Koehn et al., 2007; Wubben et al., 2010; Bannard and Callison-Burch, 2005). Xu et al. (2015) presented the task of paraphrase generation while targeting a particular writing style, specifically paraphrasing modern English into Shakespearean English, and approached it with phrase based MT.

Work on paraphrasing and summarization is often evaluated using MT evaluation measures such as BLEU. As BLEU is *precision-oriented*, complementary *recall-oriented* measures are often used as well. A prominent example is ROUGE (Lin, 2004), a family of measures used mostly for evaluation in automatic summarization: candidate summaries are scored according to the fraction of n-grams from the human references they contain.

We also utilize PINC (Chen and Dolan, 2011), a measure which rewards paraphrases for being different from their source, by introducing new n-grams. PINC is often combined with BLEU due to their complementary nature: while PINC rewards n-gram novelty, BLEU rewards similarity to the reference. The highest correlation with human judgments is achieved by the product of PINC with a sigmoid function of BLEU (Chen and Dolan, 2011).

## 3 A Parallel Sarcastic Tweets Corpus

To properly investigate our task, we collected a dataset, first of its kind, of sarcastic tweets and their non-sarcastic (honest) interpretations. This data, as well as the instructions provided for our human judges, will be made publicly available and will hopefully provide a basis for future work regarding sarcasm on Twitter. Despite the focus of the current work on the Twitter domain, we consider our task as a more general one, and hope that our discussion, observations and algorithms will be beneficial for other domains as well.

Using the Twitter API[2], we collected tweets marked with the content tag #sarcasm, posted between Januray and June of 2016. Following Tsur et al. (2010), González-Ibánez et al. (2011) and Bamman and Smith (2015), we address the problem of noisy tweets with automatic filtering: we remove all tweets not written in English, discard retweets (tweets that have been forwarded or shared) and remove tweets containing URLs or images, so that the sarcasm in the tweet regards to the text only and not to an image or a link. This results in 3000 sarcastic tweets containing text only.

In order to obtain honest interpretations for our sarcastic tweets, we used Fiverr[3] – a platform for selling and purchasing services from independent suppliers (also referred to as workers). We employed ten Fiverr workers, half of them from the field of comedy writing, and half from the field of literature paraphrasing. The chosen workers were made sure to have an active Twitter account, in order to ensure their acquaintance with social networks and with Twitter's colorful language (hashtags, common acronyms such as LOL, etc.).

---

[2]http://apiwiki.twitter.com
[3]https://www.fiverr.com

| Sarcastic Tweets | Honest Interpretations |
|---|---|
| What a great way to end my night. #sarcasm | 1. Such a bad ending to my night<br>2. Oh what a great way to ruin my night<br>3. What a horrible way to end a night<br>4. Not a good way to end the night<br>5. Well that wasn't the night I was hoping for |
| Staying up till 2:30 was a brilliant idea, very productive #sarcasm | 1. Bad idea staying up late, not very productive<br>2. It was not smart or productive for me to stay up so late<br>3. Staying up till 2:30 was not a brilliant idea, very non-productive<br>4. I need to go to bed on time<br>5. Staying up till 2:30 was completely useless |

Table 1: Examples from our parallel sarcastic tweet corpus.

We then randomly divided our tweet corpus to two batches of size 1500 each, and randomly assigned five workers to each batch. We instructed the workers to translate each sarcastic tweet into a non sarcastic utterance, while maintaining the original meaning. We encouraged the workers to use external knowledge sources (such as Google) if they came across a subject they were not familiar with, or if the sarcasm was unclear to them.

Table 1 presents two examples from our corpus. The table demonstrates the tendency of the workers to generally agree on the core meaning of the sarcastic tweets. Yet, since sarcasm is inherently vague, it is not surprising that the interpretations differ from one worker to another. For example, some workers change only one or two words from the original sarcastic tweet, while others rephrase the entire utterance. We regard this as beneficial, since it brings a natural, human variance into the task. This variance makes the evaluation of automatic sarcasm interpretation algorithms challenging, as we further discuss in the next section.

## 4 Evaluation Measures

As mentioned above, in certain cases world knowledge is mandatory in order to correctly evaluate sarcasm interpretations. For example, in the case of the second sarcastic tweet in table 1, we need to know that 2:30 is considered a late hour so that *staying up till 2:30* and *staying up late* would be considered equivalent despite the lexical difference. Furthermore, we notice that transforming a sarcastic utterance into a non sarcastic one often requires to change a small number of words. For example, a single word change in the sarcastic tweet *"How I love Mondays. #sarcasm"* leads to the non-sarcastic utterance *How I hate Mondays*.

This is not typical for MT, where usually the entire source sentence is translated to a new sentence in the target language and we would expect lexical similarity between the machine generated translation and the human reference it is compared to. This raises a doubt as to whether n-gram based MT evaluation measures such as the aforementioned are suitable for our task. We hence asses the quality of an interpretation using *automatic evaluation measures* from the tasks of MT, paraphrasing, and summarization (Section 2), and compare these measures to *human-based measures*.

**Automatic Measures** We use BLEU and ROUGE as measures of n-gram precision and recall, respectively. We report scores of ROUGE-1, ROUGE-2 and ROUGE-L (recall based on unigrams, bigrams and longest common subsequence between candidate and reference, respectively). In order to asses the n-gram novelty of interpretations (i.e, difference from the source), we report PINC and PINC∗sigmoid(BLEU) (see Section 2).

**Human judgments** We employed an additional group of five Fiverr workers and asked them to score each generated interpretations with two scores on a 1-7 scale, 7 being the best. The scores are: *adequacy*: the degree to which the interpretation captures the meaning of the original tweet; and *fluency*: how readable the interpretation is. In addition, reasoning that a high quality interpretation is one that captures the true intent of the sarcastic utterance by using words suitable to its sentiment, we ask the workers to assign the interpretation with a binary score indicating whether the sentiment presented in the interpretation agrees with the sentiment of the original sarcastic tweet.[4]

The human measures enjoy high agreement levels between the human judges. The averaged root mean squared error calculated on the test set

---

[4]For example, we consider *"Best day ever #sarcasm"* and its interpretation *"Worst day ever"* to agree on the sentiment, despite the use of opposite sentiment words.

| Sarcastic Tweet | Moses Interpretation | Neural Interpretation |
|---|---|---|
| Boy , am I glad the rain's here #sarcasm | Boy, I'm so annoyed that the rain is here | I'm not glad to go today |
| Another night of work, Oh, the joy #sarcasm | Another night of work, Ugh, unbearable | Another night, I don't like it |
| Being stuck in an airport is fun #sarcasm | Be stuck in an airport is not fun | Yay, stuck at the office again |
| You're the best. #sarcasm | You're the best | You're my best friend |

Table 2: Sarcasm interpretations generated by Moses and by the RNN.

|  | Evaluation Measure | Moses | RNN |
|---|---|---|---|
| Precision Oriented | BLEU | 62.91 | 41.05 |
| Novelty Oriented | PINC | 51.81 | 76.45 |
|  | PINC*sigmoid(BLEU) | 33.79 | 45.96 |
| Recall Oriented | ROUGE-1 | 66.44 | 42.20 |
|  | ROUGE-2 | 41.03 | 29.97 |
|  | ROUGE-l | 65.31 | 40.87 |
| Human Judgments | Fluency | 6.46 | 5.12 |
|  | Adequacy | 2.54 | 2.08 |
|  | % correct sentiment | 28.84 | 17.93 |

Table 3: Development data results for MT models.

across all pairs of judges and across the various algorithms we experiment with are: 1.44 for fluency and 1.15 for adequacy. For sentiment scores the averaged agreement at the same setup is 93.2%.

## 5 Sarcasm Interpretations as MT

As our task is about the generation of one English sentence given another, a natural starting point is treating it as monolingual MT. We hence begin with utilizing two widely used MT systems, representing two different approaches: Phrase Based MT vs. Neural MT. We then analyze the performance of these two systems, and based on our conclusions we design our SIGN model.

**Phrase Based MT** We employ Moses[5], using word alignments extracted by GIZA++ (Och and Ney, 2003) and symmetrized with the grow-diag-final strategy. We use phrases of up to 8 words to build our phrase table, and do not filter sentences according to length since tweets contain at most 140 characters. We employ the KenLM algorithm (Heafield, 2011) for language modeling, and train it on the non-sarcastic tweet interpretations (the target side of the parallel corpus).

**Neural Machine Translation** We use Ground-Hog, a publicly available implementation of an RNN encoder-decoder, with LSTM hidden states.[6] Our encoder and decoder contain 250 hidden units each. We use the minibatch stochastic gradient

---

[5]http://www.statmt.org/moses

[6]https://github.com/lisa-groundhog/GroundHog

descent (SGD) algorithm together with Adadelta (Zeiler, 2012) to train each model, where each SGD update is computed using a minibatch of 16 utterances. Following Sutskever et al. (2014), we use beam search for test time decoding. Henceforth we refer to this system as *RNN*.

**Performance Analysis** We divide our corpus into training, development and test sets of sizes 2400, 300 and 300 respectively. We train Moses and the RNN on the training set and tune their parameters on the development set. Table 3 presents *development data* results, as these are preliminary experiments that aim to asses the compatibility of MT algorithms to our task.

Moses scores much higher in terms of BLEU and ROUGE, meaning that compared to the RNN its interpretations capture more n-grams appearing in the human references while maintaining high precision. The RNN outscores Moses in terms of PINC and PINC*sigmoid(BLEU), meaning that its interpretations are more novel, in terms of n-grams. This alone might not be a negative trait; However, according to human judgments Moses performs better in terms of fluency, adequacy and sentiment, and so the novelty of the RNN's interpretations does not necessarily contribute to their quality, and even possibly reduces it.

Table 2 illustrates several examples of the interpretations generated by both Moses and the RNN. While the interpretations generated by the RNN are readable, they generally do not maintain the meaning of the original tweet. We believe that this is the result of the neural network overfitting the training set, despite regularization and dropout layers, probably due to the relatively small training set size. In light of these results when we experiment with the SIGN algorithm (Section 7), we employ Moses as its MT component.

The final example of Table 2 is representative of cases where both Moses and the RNN fail to capture the sarcastic sense of the tweet, incorrectly interpreting it or leaving it unchanged. In order to deal with such cases, we wish to utilize a property typical of sarcastic language. Sarcasm is mostly

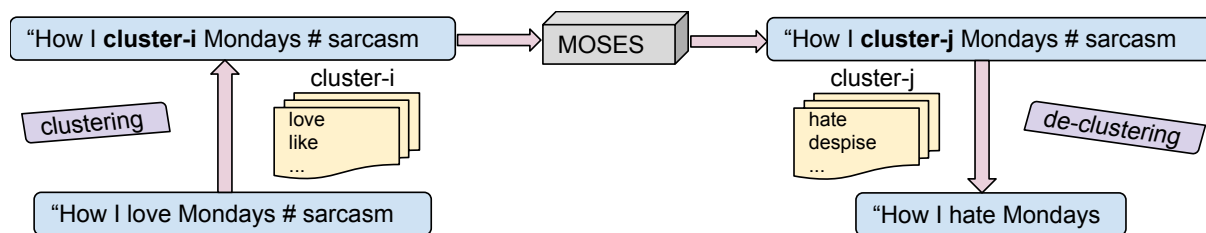

Figure 1: An illustration of the application of SIGN to the tweet *"How I love Mondays # sarcasm"*.

| Positive Clusters | merit, wonder, props, praise, congratulations.. | patience, dignity, truth, chivalry, rationality... |
|---|---|---|
| Negative Clusters | hideous, horrible, nasty, obnoxious, scary, pathetic... | shame, sadness, sorrow, fear, disappointment, regret, danger... |

Table 4: Examples of two positive and two negative clusters created by the SIGN algorithm.

used to convey a certain emotion by using strong sentiment words that express the exact opposite of their literal meaning. Hence, many sarcastic utterances can be correctly interpreted by keeping most of their words, replacing only sentiment words with expressions of the opposite sentiment. For example, the sarcasm in the utterance *"You're the best. #sarcasm"* is hidden in *best*, a word of a strong positive sentiment. If we transform this word into a word of the opposite sentiment, such as *worst*, then we get a non-sarcastic utterance with the correct sentiment.

We next present the *Sarcasm SIGN* (*Sarcasm Sentimental Interpretation GeNerator*), an algorithm which capitalizes on sentiment words in order to produce accurate interpretations.

## 6 The Sarcasm SIGN Algorithm

SIGN (Figure 1) targets sentiment words in sarcastic utterances. First, it clusters sentiment words according to semantic relatedness. Then, each sentiment word is replaced with its cluster [7] and the transformed data is fed into an MT system (Moses in this work), at both its training and test phases. Consequently, at test time the MT system outputs non-sarcastic utterances with clusters replacing sentiment words. Finally, SIGN performs a de-clustering process on these MT outputs, replacing sentiment clusters with suitable words.

In order to detect the sentiment of words, we turn to SentiWordNet (Esuli and Sebastiani, 2006), a lexical resource based on WordNet (Miller et al., 1990; Kilgarriff and Fellbaum, 2000). Using SentiWordNet's positivity and negativity scores, we collect from our training data a set of distinctly positive words ($\sim$ 70) and a set of distinctly negative words ($\sim$ 160).[8] We then utilize the pre-trained dependency-based word embeddings of Levy and Goldberg (2014)[9] and cluster each set using the k-means algorithm with $L2$ distance. We aim to have ten words on average in each cluster, and so the positive set is clustered into 7 clusters, and the negative set into 16 clusters. Table 4 presents examples from our clusters.

Upon receiving a sarcastic tweet, at both training and test, SIGN searches it for sentiment words according to the positive and negative sets. If such a word is found, it is replaced with its cluster. For example, given the sentence *"How I love Mondays. #sarcasm"*, *love* will be recognized as a positive sentiment word, and the sarcastic tweet will become: *"How I cluster-i Mondays. #sarcasm"* where $i$ is the cluster number of the word *love*.

During training, this process is also applied to the non-sarcastic references. And so, if one such reference is *"I dislike Mondays."*, then *dislike* will be identified and the reference will become *"I cluster-j Mondays."*, where $j$ is the cluster number of the word *dislike*. Moses is then trained on these new representations of the corpus, using the exact same setup as before. This training process produces a mapping between positive and negative clusters, and outputs sarcastic interpretations with clustered sentiment words (e.g, *"I cluster-j Mondays."*). At test time, after Moses generates an utterance containing clusters, a de-clustering pro-

---

[7]This means that we replace a *word* with *cluster-j* where $j$ is the number of the cluster to which the *word* belongs.

[8]The scores are in the [0,1] range. We set the threshold of 0.6 for both distinctly positive and distinctly negative words.

[9]https://levyomer.wordpress.com/2014/04/25/dependency-based-word-embeddings/. We choose these embeddings since they are believed to better capture the relations between a word and its context, having been trained on dependency-parsed sentences.

|  | Evaluation Measure | Moses | SIGN-centroid | SIGN-context | SIGN-oracle |
|---|---|---|---|---|---|
| Precision Oriented | BLEU | 65.24 | 63.52 | **66.96** | 67.49 |
| Novelty Oriented | PINC | 45.92 | **47.11** | 46.65 | 46.10 |
|  | PINC∗sigmoid(BLEU) | 30.21 | 30.79 | **31.13** | 30.54 |
| Recall Oriented | ROUGE-1 | **70.26** | 68.43 | 69.67 | 70.34 |
|  | ROUGE-2 | **42.18** | 40.34 | 40.96 | 42.81 |
|  | ROUGE-l | 69.82 | 68.24 | **69.98** | 70.01 |

Table 5: Test data results with automatic evaluation measures.

cess takes place: the clusters are replaced with the appropriate sentiment words.

We experiment with several de-clustering approaches: **(1) SIGN-centroid:** the chosen sentiment word will be the one closest to the centroid of cluster j. For example in the tweet *"I cluster-j Mondays."*, the sentiment word closest to the centroid of cluster j will be chosen; **(2) SIGN-context:** the cluster is replaced with its word that has the highest average Pointwise Mutual Information (PMI) with the words in a symmetric context window of size 3 around the cluster's location in the output. For example, for *"I cluster-j Mondays."*, the sentiment word from cluster j which has the highest average PMI with the words in {'I','Mondays'} will be chosen. The PMI values are computed on the training data; and **(3) SIGN-Oracle:** an upper bound where a person manually chooses the most suitable word from the cluster.

We expect this process to improve the quality of sarcasm interpretations in two aspects. First, as mentioned earlier, sarcastic tweets often differ from their non sarcastic interpretations in as little as one sentiment word. SIGN should help highlight the sentiment words most in need of interpretation. Second, under the pre-processing SIGN performs to the input examples of Moses, the latter is inclined to learn a mapping from positive to negative clusters, and vice versa. This is likely to encourage the Moses output to generate outputs of the same sentiment as the original sarcastic tweet, but with honest sentiment words. For example, if the sarcastic tweet expresses a negative sentiment with strong positive words, the non-sarcastic interpretation will express this negative sentiment with negative words, thus stripping away the sarcasm.

## 7 Experiments and Results

We experiment with SIGN and the Moses and RNN baselines at the same setup of section 5. We report test set results for automatic and human measures, in Tables 5 and 6 respectively. As in

|  | Fluency | Adequacy | % correct sentiment | % changed |
|---|---|---|---|---|
| Moses | **6.67** | 2.55 | 25.7 | 42.3 |
| SIGN-Centroid | 6.38 | 3.23* | 42.2* | 67.4 |
| SIGN-Context | 6.66 | **3.61*** | **46.2*** | **68.5** |
| SIGN-Oracle | 6.69 | 3.67* | 46.8* | 68.8 |

Table 6: Test set results with human measures. *%changed* provides the fraction of tweets that were changed during interpretation (i.e. the tweet and its interpretation are not identical). In cases where one of our models presents significant improvement over Moses, the results are decorated with a star. Statistical significance is tested with the paired t-test for fluency and adequacy, and with the McNemar paired test for labeling disagreements (Gillick and Cox, 1989) for *% correct sentiment*, in both cases with $p < 0.05$.

the development data experiments (Table 3), the RNN presents critically low adequacy scores of 2.11 across the entire test set and of 1.89 in cases where the interpretation and the tweet differ. This, along with its low fluency scores (5.74 and 5.43 respectively) and its very low BLEU and ROUGE scores make us deem this model immature for our task and dataset, hence we exclude it from this section's tables and do not discuss it further.

In terms of automatic evaluation (Table 5), SIGN and Moses do not perform significantly different. When it comes to human evaluation (Table 6) however, SIGN-context presents substantial gains. While for fluency Moses and SIGN-context perform similarly, SIGN-context performs much better in terms of adequacy and the percentage of tweets with the correct sentiment. The differences are substantial as well as statistically significant: adequacy of 3.61 for SIGN-context compared to 2.55 of Moses, and correct sentiment for 46.2% of the SIGN interpretations, compared to only 25.7% of the Moses interpretations.

Table 6 further provides an initial explanation to the improvement of SIGN over Moses: Moses tends to keep interpretations identical to the origi-

nal sarcastic tweet, altering them in only 42.3% of the cases, [10] while SIGN-context's interpretations differ from the original sarcastic tweet in 68.5% of the cases, which comes closer to the 73.8% in the gold standard human interpretations. If for each of the algorithms we only regard to interpretations that differ from the original sarcastic tweet, the differences between the models are less substantial. Nonetheless, SIGN-context still presents improvement by correctly changing sentiment in 67.5% of the cases compared to 60.8% for Moses.

Both tables consistently show that the context-based selection strategy of SIGN outperforms the centroid alternative. This makes sense as, being context-ignorant, SIGN-centroid might produce non-fluent or inadequate interpretations for a given context. For example, the tweet *"Also gotta move a piano as well. joy #sarcasm"* is changed to *"Also gotta move a piano as well. bummer"* by SIGN-context, while SIGN-centroid changes it to the less appropriate *"Also gotta move a piano as well. boring"*. Nonetheless, even this naive de-clustering approach substantially improves adequacy and sentiment accuracy over Moses.

Finally, comparison to SIGN-oracle reveals that the context selection strategy is not far from human performance with respect to both automatic and human evaluation measures. Still, some gain can be achieved, especially for the human measures on tweets that were changed at interpretation. This indicates that SIGN can improve mostly through a better clustering of sentiment words, rather than through a better selection strategy.

# 8 Discussion and Future Work

The performance gap between Moses and SIGN may stem from the difference in their optimization criteria. Moses aims to optimize the BLEU score and given the overall lexical similarity between the original tweets and their interpretations, it therefore tends to keep them identical. SIGN, in contrast, targets sentiment words and changes them frequently. Consequently, we do not observe substantial differences between the algorithms in the automatic measures that are mostly based on n-gram differences between the source and the interpretation. Likewise, the human fluency measure that accounts for the readability of the interpretation is not seriously affected by the translation process. When it comes to the human adequacy and

---

[10] We elaborate on this in section 8.

sentiment measures, which account for the understanding of the tweet's meaning, SIGN reveals its power and demonstrates much better performance compared to Moses.

To further understand the relationship between the automatic and the human based measures we computed the Pearson correlations for each pair of (automatic, human) measures. We observe that all correlation values are low (up to 0.12 for fluency, 0.13-0.18 for sentiment and 0.19-0.24 for adequacy). Moreover, for fluency the correlation values are insignificant (using a correlation significance t-test with $p = 0.05$). We believe this indicates that these automatic measures do not provide appropriate evaluation for our task. Designing automatic measures is hence left for future research.

A qualitative analysis reveals that as expected, SIGN-context performs well on sarcastic tweets with clear sentiment words, transforming expressions such as *"Audits are a blast to do #sarcasm"* and *"Being stuck in an airport is fun #sarcasm"* into *"Audits are a bummer to do"* and *"Being stuck in an airport is boring"*, respectively. Even when there are several sentiment words and not all of them require a change, e.g. in *"Constantly being irritated, anxious and depressed is a great feeling! #sarcasm"*, SIGN-context produces the adequate interpretation: *"Constantly being irritated, anxious and depressed is a terrible feeling"*.

In some cases, however, SIGN struggles with producing correct interpretations. For example, the tweet *"Can you imagine if Lebron had help? #sarcasm"*, was left unchanged by all SIGN models (including the upper bound oracle). Notice that the sarcasm in this tweet is not expressed by specific sentiment words. Moreover, world knowledge (who Lebron is, what kind of help he requires) is crucial in order to comprehend and interpret this sarcastic utterance. In addition, phrases that express sentiment without explicit sentiment words, for example *"can't wait"* in the tweet *"Can't wait until tomorrow #sarcasm"*, diminish SIGN's advantage. Further Improving SIGN so that it properly interprets such tweets is a major future direction.

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
