# Peer review of "Sarcasm SIGN: Interpreting Sarcasm with Sentiment Based Monolingual Machine Translation"

_ACL 2017 — decision unknown_

[Official Review · Reviewer 1 · rating 3 · confidence 3]
soundness 4 · originality 3 · clarity 4 · impact 3 · substance 4 · appropriateness 5 · meaningful comparison 4 · presentation format Poster

- Summary: 

The paper introduces a new dataset for a sarcasm interpretation task
and a system (called Sarcasm SIGN) based on machine translation framework
Moses. The new dataset was collected from 3000 sarcastic tweets (with hashtag
`#sarcasm) and 5 interpretations for each from humans. The Sarcasm SIGN is
built
based on Moses by replacing sentimental words by their corresponding clusters
on the source side (sarcasm) and then de-cluster their translations on the
target side (non-sarcasm). Sarcasm SIGN performs on par with Moses on the MT
evaluation metrics, but outperforms Moses in terms of fluency and adequacy. 

- Strengths:

the paper is well written

the dataset is collected in a proper manner

the experiments are carefully done and the analysis is sound.

- Weaknesses:

lack statistics of the datsets (e.g. average length, vocabulary size)

the baseline (Moses) is not proper because of the small size of the dataset

the assumption "sarcastic tweets often differ from their non sarcastic
interpretations in as little as one sentiment word" is not supported by the
data. 

- General Discussion: This discussion gives more details about the weaknesses
of the paper. 

Half of the paper is about the new dataset for sarcasm interpretation.
However, the paper doesn't show important information about the dataset such as
average length, vocabulary size. More importantly, the paper doesn't show any
statistical evidence to support their method of focusing on sentimental words. 

Because the dataset is small (only 3000 tweets), I guess that many words are
rare. Therefore, Moses alone is not a proper baseline. A proper baseline should
be a MT system that can handle rare words very well. In fact, using
clustering and declustering (as in Sarcasm SIGN) is a way to handle rare words.

Sarcasm SIGN is built based on the assumption that "sarcastic tweets often
differ from their non sarcastic interpretations in as little as one sentiment
word". Table 1 however strongly disagrees with this assumption: the human
interpretations are often different from the tweets at not only sentimental
words. I thus strongly suggest the authors to give statistical evidence from
the dataset that supports their assumption. Otherwise, the whole idea of
Sarcasm SIGN is just a hack.

--------------------------------------------------------------

I have read the authors' response. I don't change my decision because of the
following reasons: 

- the authors wrote that "the Fiverr workers might not take this strategy": to
me it is not the spirit of corpus-based NLP. A model must be built to fit given
data, not that the data must follow some assumption that the model is built on.

- the authors wrote that "the BLEU scores of Moses and SIGN are above 60, which
is generally considered decent in the MT literature": to me the number 60
doesn't 
show anything at all because the sentences in the dataset are very short. And
that,
if we look at table 6, %changed of Moses is only 42%, meaning that even more
than half of the time translation is simply copying, the BLUE score is more
than 60.

- "While higher scores might be achieved with MT systems that explicitly
address rare words, these systems don't focus on sentiment words": it's true,
but I was wondering whether sentiment words are rare in the corpus. If they
are, those MT systems should obviously handle them (in addition to other rare
words).

[Official Review · Reviewer 2 · rating 3 · confidence 3]
soundness 4 · originality 3 · clarity 5 · impact 3 · substance 4 · appropriateness 5 · meaningful comparison 4 · presentation format Poster

- Strengths:

(1) A new dataset would be useful for other researchers in this area

(2) An algorithm with sentiment words based machine translation is proposed to
interpret sarcasm tweets.

- Weaknesses:

(1) Do not provide detailed statistics of constructed dataset.

(2) Integrating sentiment word clustering with machine translation techniques
only is simple and straightforward, novelty may be a challenging issue. 

- General Discussion:

Overall, this paper is well written. The experiments are conducted carefully
and the analysis is reasonable. 

I offer some comments as follows.
(1) According to data collection process, each tweet should be annotated
five times. How to determine which one is regarded as gold standard for measure
performance?

(2) The MT technique (Moses) is well known, but it may not be a good
baseline. Another MT technique (RNN) should be put together for comparison.   

(3) Differ from most work focuses on sarcasm detection. The research topic
is interesting. It attempts to interpret sarcasm for reflecting semantics.

[Official Review · Reviewer 3 · rating 4 · confidence 4]
soundness 4 · originality 3 · clarity 5 · impact 3 · substance 4 · appropriateness 5 · meaningful comparison 4 · presentation format Poster

This paper focuses on interpreting sarcasm written in Twitter identifying
sentiment words and then using a machine translation engine to find an
equivalent not sarcastic tweet. 

EDIT: Thank you for your answers, I appreaciate it. I added one line commenting
about it.

- Strengths:

Among the positive aspects of your work, I would like to mention the parallel
corpus you presented. I think it will be very useful for other researchers in
the area for identifying and interpreting sarcasm in social media. An important
contribution is also the attempt to evaluate the parallel corpora using
existing measures such as the ones used in MT tasks. But also because you used
human judgements to evaluate the corpora in 3 aspects: fluency, adequacy and
equivalent sentiment.

- Room for improvement:

Tackling the problem of interpretation as a monolingual machine translations
task is interesting, while I do appreciate the intent to compare the MT with
two architectures, I think that due the relatively small dataset (needed for
RNN) used it was predictable that the “Neural interpretation” is performing
worse than “moses interpretation”. You came to the same conclusion after
seeing the results in Table3. In addition to comparing with this architecture,
I would've liked to see other configuration of the MT used with moses. Or at
least, you should provide some explanation of why you use the configuration
described in lines 433 through 442; to me this choice is not justified. 
  - thank you for your response, I understand it is difficult to write down all
the details but I hope you include a line with some of your answer in the
paper, I believe this could add valuable information.

When you presented SING, it is clear that you evaluate some of its components
beforehand, i.e. the MT. But other important components are not evaluated,
particularly, the clustering you used of positive and negative words. While you
did said you used k-means as a clustering algorithm it is not clear to me why
you wanted to create clusters with 10 words. Why not test with other number of
k, instead of 7 and 16, for positive and negative words respectively. Also you
could try another algorithm beside kmeans, for instance, the star clustering
algorithm (Aslam et al. 2004), that do not require a k parameter. 
   - thanks for clarifying.

You say that SIGN searches the tweet for sentiment words if it found one it
changes it for the cluster ID that contain that word. I am assuming that there
is not a limit for the number of sentiment words found, and the MT decides by
itself how many sentiment words to change. For example, for the tweet provided
in Section 9: “Constantly being irritated, anxious and depressed is a great
feeling” the clustering stage of SIGN should do something like “Constantly
being cluster-i, cluster-j and cluster-k is a cluster-h feeling”, Is that
correct? If not, please explain what SIGN do.
    - Thanks for clarifying

- Minor comments:

In line 704, section 7, you said: “SIGN-context’s interpretations differ
from the original sarcastic tweet in 68.5% of the cases, which come closer to
the 73.8% in the gold standard human interpretations.” This means that 25% of
the human interpretations are the same as the original tweet? Do you have any
idea why is that?

In section 6, line 539 you could eliminate the footnote 7 by adding “its
cluster ID” or “its cluster number”.

References:
Aslam, Javed A., Pelekhov, Ekaterina, and Rus, Daniela. "The star clustering
algorithm for static and dynamic information organization.." Journal of Graph
Algorithms and Applications 8.1 (2004): 95-129. <http://eudml.org/doc/51529>.